# Trans-(−)-Kusunokinin: A Potential Anticancer Lignan Compound against HER2 in Breast Cancer Cell Lines?

**DOI:** 10.3390/molecules26154537

**Published:** 2021-07-27

**Authors:** Thidarath Rattanaburee, Tanotnon Tanawattanasuntorn, Tienthong Thongpanchang, Varomyalin Tipmanee, Potchanapond Graidist

**Affiliations:** 1Department of Biomedical Sciences and Biomedical Engineering, Faculty of Medicine, Prince of Songkla University, Songkhla 90110, Thailand; thi325@gmail.com (T.R.); thawaii39@gmail.com (T.T.); 2Department of Chemistry and Center of Excellence for Innovation in Chemistry (PERCH-CIC), Faculty of Science, Mahidol University, Bangkok 10400, Thailand; tienthong.tho@mahidol.ac.th

**Keywords:** kusunokinin, HER2, breast cancer, neratinib

## Abstract

Trans-(−)-kusunokinin, an anticancer compound, binds CSF1R with low affinity in breast cancer cells. Therefore, finding an additional possible target of trans-(−)-kusunokinin remains of importance for further development. Here, a computational study was completed followed by indirect proof of specific target proteins using small interfering RNA (siRNA). Ten proteins in breast cancer were selected for molecular docking and molecular dynamics simulation. A preferred active form in racemic trans-(±)-kusunokinin was trans-(−)-kusunokinin, which had stronger binding energy on HER2 trans-(+)-kusunokinin; however, it was weaker than the designed HER inhibitors (03Q and neratinib). Predictively, trans-(−)-kusunokinin bound HER2 similarly to a reversible HER2 inhibitor. We then verified the action of (±)-kusunokinin compared with neratinibon breast cancer cells (MCF-7). (±)-Kusunokinin exhibited less cytotoxicity on normal L-929 and MCF-7 than neratinib. (±)-Kusunokinin and neratinib had stronger inhibited cell proliferation than siRNA-HER2. Moreover, (±)-kusunokinin decreased Ras, ERK, CyclinB1, CyclinD and CDK1. Meanwhile, neratinib downregulated HER, MEK1, ERK, c-Myc, CyclinB1, CyclinD and CDK1. Knocking down HER2 downregulated only HER2. siRNA-HER2 combination with (±)-kusunokinin suppressed HER2, c-Myc, CyclinB1, CyclinD and CDK1. On the other hand, siRNA-HER2 combination with neratinib increased HER2, MEK1, ERK, c-Myc, CyclinB1, CyclinD and CDK1 to normal levels. We conclude that trans-(±)-kusunokinin may bind HER2 with low affinity and had a different action from neratinib.

## 1. Introduction

Breast cancer is the most common type of cancer worldwide. In 2020, female breast cancer was estimated at 2.26 million new cases. The new death cases of female breast cancer accounted for almost 0.68 million and approximately 6.9% of all cancer sites [1]. The treatments for breast cancer are surgery, medical oncology (such as chemotherapy, endocrine therapy and targeted therapy) and radiation [2]. According to gene expression profiles, breast cancers are classified into four types, including luminal A (estrogen receptor positive (ER+), progesterone receptor positive (PR+), human epidermal growth factor receptor 2 negative (HER2-)), luminal B (ER+, PR+, HER2+), non-luminal with HER2+ (ER-, PR-, HER2+) and basal-like or triple negative (ER-, PR-, HER2-) [3]. The treatment for luminal A patients is surgery and hormonal therapy [4]. Surgery, adjuvant chemotherapy and radiotherapy are suitable for luminal B and non-luminal with HER2+ breast cancers [5]. The PARP inhibitor, platinum chemotherapy and immunotherapy are used for triple-negative breast cancer [6]. Breast cancer patients who represent ER+/PR+ and HER2+ have a specific drug target, including tamoxifen [7] and trastuzumab (Herceptin™) [8], respectively. ER, PR and HER2 relate to breast cancer growth. HER2 is a protein in the epidermal growth factor receptor (EGFR) family. This family consists of four protein members including ErbB1 (EGFR or HER1), ErbB2 (HER2), ErbB3 (HER3) and ErbB4 (HER4) [9]. In addition, four proteins in the cancer cell growth pathway are targeted for therapeutic purposes in breast cancer, including cyclin-dependent kinase 4 and 6 (CDK4/6) [10,11], poly (ADP-ribose) polymerase (PARP) [12] and phosphoinositide-3-kinase (PI3K) [13]. CDK4/6 inhibitors are palbociclib, ribociclib and abemaciclib [14]. PARP relates to DNA repair, genomic stability and programmed cell death. Olaparib and talazoparib are PARP inhibitors [15]. PI3K inhibitor is piqray (alpelisib) [16]. The Ras related protein RalA (RalA) is important in tumor growth and metastasis. RBC8 is a small molecule that binds to a site on the GDP-bound form of RalA [17].

Trans-(−)-kusunokinin, a lignan compound, isolated from *Piper nigrum* inhibits breast cancer cell growth, and induces cell cycle blockage, apoptosis and cell cycle arrest at the Gap 2/mitotic (G2/M) phase [18]. The synthetic compound of trans-kusunokinin consists of two forms, including trans-(−)-kusunokinin and trans-(+)-kusunokinin. This synthetic compound exhibits similar cytotoxic activity on breast and colon cancer, the same as the extract. The synthetic trans-(±)-kusunokinin is found to suppress topoisomerase II, STAT3, CyclinD1 and p21 on breast cancer and cholangiocarcinoma cells [19]. For the in vivo study, trans-(−)-kusunokinin decreases tumor growth and migration without side effects on blood parameters and the clinical chemistry of renal and liver function [20]. Interestingly, the synthetic (±)-kusunokinin inhibits the proliferation of breast cancer cells through the binding and the suppression of CSF1R, which consequently decreases AKT and the downstream proteins in cell proliferation (CyclinD1 and CDK) [21]. Using computational analysis, trans-(−)-kusunokinin binds AKR1B1, and the upstream molecules of the AKT signaling protein in migration [22]. In this recent study, we performed a computer simulation and an in vitro study in order to determine the target protein of racemic trans-kusunokinin in breast cancer.

## 2. Results

### 2.1. Molecular Docking of Trans-(−)-Kusunokinin and Trans-(+)-Kusunokinin with Breast Cancer Related-Candidate Proteins

The docking scores of both trans-(−)-kusunokinin and trans-(+)-kusunokinin against 10 breast cancer associated proteins are shown in Table 1. All PDB codes of these 10 proteins and PubChem CID of all given inhibitors were summarized in the Appendix A (Appendix A). The docking score was given as an effective binding energy (ΔGbind) in kcal/mol although it may differ from the true binding energy. HER2 became of interest as the first prioritized target as trans-(±)-kusunokinin because HER2 was suggested as the challenging targets for breast cancer therapy [23], due to its drug resistance, cross-resistance to other HER2-targeted drugs [24,25], and drug toxicity [26,27]. These preliminary results and information encouraged further simulation study of trans-(−)-kusunokinin and trans-(+)-kusunokinin with HER2, along with the experiment of synthetic inhibitors and neratinib (HKI-272). Molecular docking could, however, only roughly guide the potential binding site of the drug towards an interested protein. MD simulation was thus undertaken in order to improve the prediction of the relative binding energy of the compounds compared to the known inhibitors, and the inspection of binding behaviors with an effect of temperature, pressure and an aqueous environment.

### 2.2. Atomistic Features of Trans-(−)-Kusunokinin and Trans-(+)-Kusunokinin and HER2

This study consisted of five MD simulations: four MD trajectories of ligand-protein complexes (trans-(−)-kusunokinin, trans-(+)-kusunokinin, synthetic inhibitor 03q and neratinib) and a simulation of the HER2 protein. The mentioned ligands are illustrated in Figure 1.

Structural analysis was performed based on two plots: RMSD and RMSF plots (Figure 2). The RMSD plot denoted the stability of the MD simulation after 50 ns of NPT simulation time since the RMSD values in all five simulations constantly fluctuated (Figure 2A). Hence, taking the last 50 ns simulation was valid in terms of both structure and binding analysis. The plot of RMSD from the beginning of the simulation is shown in Appendix A (Appendix A). In addition, the RMSF pattern of the ligand-bound HER2 was illustrated together with ligand-free HER, Figure 2B; therefore, the effect of a bound ligand on flexibility could be easily detected. The difference in flexibility patterns of kusunokinin compared to ligand free HER2 was observed from the following amino acid residues: 50th to 65th, 80th to 90th, and 100th to 110th, respectively. Nevertheless, the flexibility pattern of kusunokinin only shared similarities with neratinib at some points, from the residues 50th to 65th residues, meanwhile other flexibility parts were clearly different.

In Figure 3, the compound interacted towards HER2 protein via hydrophobic interaction since the cavity of the HER2 binding site was mostly composed of nonpolar amino acids, namely isoleucine, leucine, valine, methionine, alanine and phenylalanine, respectively. Trans-(−)-kusunokinin and trans-(+)-kusunokinin had bound to the same HER2 site, using hydrophobic interaction and π-π stacking due to Phenylalaine159 (Phe159). However, the hydrogen bond between oxygen atoms in the lactone ring and lysine48 (Lys48) could be a selective key to facilitate better HER2 binding of trans-(−)-kusunokinin, as shown in Figure 3A,B, respectively. Phe159 was also found to contribute interaction with the aromatic ring of all compounds, except neratinib, as shown in Figure 3A–C. The betting binding affinity (lower binding energy) from the synthetic 03q likely came from more additional hydrogen bonds, such as Lysine48 (Lys48) and arginine144 (Arg144) in the case of 03q (Figure 3C). Furthermore, our simulation showed that the neratinib binding site consisted of cysteine100 (Cys100), methionine96 (Met96), leucine21 (Leu21) and threonine (Thr93), Figure 3D. This Cys100 could justify our binding prediction as it was a vital key residue in drug-HER2 interaction [28].

The relative binding energy was then evaluated from MM/GBSA and MM/PBSA based on the MD trajectory. To be noted, the relative binding energy was just able to predict the relative binding tendency of the ligand to the target protein, not to identify whether the compound can truly bind the studied target. Rather, in our study, we could predict the relative possibility of synthetic (±)-kusunokinin for HER2 binding with respect to the known inhibitors. The relative HER2 binding energies of trans-(−)-kusunokinin and trans-(+)-kusunokinin were then acquired, whereas neratinib and synthetic 03q were taken as the reference binding energy (Table 2). From Table 2, lower binding energies (better binding affinity) were observed from neratinib and synthetic 03q than kusunokinin. This was not surprising due to the fact that both inhibitors were selectively designed for HER2 specific binding. The calculations suggest that trans-(−)-kusunokinin to trans-(+)-kusunokinin would not compete well with neratinib and synthetic 03q in terms of HER2 binding. 

### 2.3. Cytotoxicity Effects of Synthetic (±)-Kusunokinin on Breast Cancer and Normal Cells

In this in vitro experiment, MCF-7, MCF-12A and L-929 cells were treated with synthetic (±)-kusunokinin and HER2 known inhibitor (neratinib) in order to verify the effect of both compounds compared with computational simulating data. Synthetic (±)-kusunokinin in racemic form was used in this study; therefore, the results are from both structures, trans-(+)-kusunokinin and trans-(−)-kusunokinin. Results showed that synthetic (±)-kusunokinin had a stronger IC_50_ value on MCF-7 (4.45 ± 0.80 µM) than L-929 (7.39 ± 1.22 µM) cells. Meanwhile, neratinib had stronger inhibited MCF-7, MCF-12A and L-929 cells than synthetic (±)-kusunokinin. Moreover, neratinib showed stronger cytotoxicity on L-929 than MCF-7 cells. These results indicate that synthetic (±)-kusunokinin exhibited less cytotoxicity than neratinib on breast cancer, normal epithelial breast and mouse fibroblast cells (Figure 4).

### 2.4. Inhibitory Effect of Synthetic (±)-Kusunokinin on Breast Cancer Cell Proliferation

To determine the inhibitory effect of synthetic (±)-kusunokinin and compare it with neratinib and siRNA-HER2, MCF-7 cells were treated with an IC_50_ value of the compound, siRNA-HER2 or a combination and incubated for 48 h. Cell viability was stained with trypan blue dye solution and calculated as the percentage of inhibition. Results showed that synthetic (±)-kusunokinin inhibited cell growth more than neratinib, 100 nM siRNA-HER2 and combinations. The combination of synthetic (±)-kusunokinin and neratinib did not show a synergistic effect. Meanwhile, knocking down HER2 along with synthetic (±)-kusunokinin treatment exhibited a percentage of inhibition lower than synthetic (±)-kusunokinin alone, but higher than siRNA-HER2. Moreover, the combination of siRNA-HER2 with neratinib showed a similar result with siRNA-HER2 alone (Figure 5).

### 2.5. Synthetic (±)-Kusunokinin Inhibited Breast Cancer Cell Proliferation through the Suppression of RAS and ERK

As suggested from the molecular simulation results, HER2 was an examined target protein in this study. HER2 and its downstream proteins (Ras, MEK1, AKT and ERK) related to cell proliferation were evaluated. The protein levels of HER2, Ras, MEK1, AKT and ERK were measured by Western blot analysis (Figure 6A). We found that HER2 was inhibited by all treatments, except a synthetic (±)-kusunokinin. The combination of neratinib and siRNA-HER2 caused the HER2 signal to be close to the normal level of the control groups (lipofectamine and siRNA-luciferase), whereas the synthetic (±)-kusunokinin combination with siRNA-HER2 did not increase HER2 protein (Figure 6B). Synthetic (±)-kusunokinin decreased RAS and ERK, meanwhile neratinib decreased MEK1 and ERK at 48 h after treatment. RAS was also decreased in the treatments of the synthetic (±)-kusunokinin combination with neratinib. Surprisingly, MEK1 and ERK proteins were increased, the same as in the control groups when MCF-7 cells were treated with the neratinib combination with siRNA-HER2. In addition, a combination of synthetic (±)-kusunokinin and siRNA-HER2 induced AKT and ERK levels close to the normal level (Figure 6B–F). These results suggest that synthetic (±)-kusunokinin inhibited cell proliferation through the reduction of RAS and ERK, which is different from neratinib because it suppressed the HER2, MEK1 and ERK proteins.

### 2.6. Synthetic (±)-Kusunokinin Decreased CyclinB1, CyclinD1 and CDK1; Down-Stream Proteins of HER2

To determine the inhibition of breast cancer cell proliferation by synthetic (±)-kusunokinin, four final proteins of the HER2 pathway were examined, including c-Myc, CyclinB1, CyclinD1 and CDK1 using Western blot analysis (Figure 7A). We found that neratinib, the combination of neratinib and synthetic (±)-kusunokinin, and the combination of siRNA and synthetic (±)-kusunokinin decreased c-Myc, CyclinB1, CyclinD1 and CDK1. Treatment with synthetic (±)-kusunokinin alone decreased CyclinB1, CyclinD1 and CDK1. Interestingly, silencing HER2 did not cause downregulation on cell proliferation proteins. The combination of neratinib and siRNA-HER2 upregulated c-Myc, CyclinB1, CyclinD1 and CDK1 to the normal baseline, the same as in the control groups (Figure 7B–E). Taken together, these results suggest that synthetic (±)-kusunokinin could function in the absence of HER2, which was different from neratinib, a HER2 known inhibitor.

## 3. Discussion

In our previous work, trans-(−)-kusunokokinin was a potential compound for cancer treatment. However, many points remain unclear, especially verified target proteins. To develop this compound to be a targeted drug for breast cancer, specific target identification was necessary for three reasons: (1) breast cancer comprises many receptors that relate to cancer treatment, (2) (−)-kusunokinin did not show strong binding with CSF1R and AKR1B1, and (3) synthetic (±)-kusunokinin contained two forms. Therefore, finding other target proteins from breast cancer associated proteins was the main objective of this work. 

Breast cancer associated receptor proteins are ER [29], PR [30], HER1 [31], HER2 [32] and HER4 [33]. In addition, targeted proteins for breast cancer therapy are also CDK4 [10], CDK6 [11], PARP [12], PI3K [13] and RAL [34]. In this current work, the specific binding of trans-(−) and trans-(+)-kusunokinin were examined with these 10 proteins via computational prediction. Computational modeling suggested that trans-(−)-kusunokinin and trans-(+)-kusunokinin bound HER2 at the same pocket as known HER inhibitors, namely the synthetic inhibitor 03q [35] and neratinib (HKI-272) [23]. 

The molecular docking approach has many limitations in its representation of atomistic information for drug-protein behaviors, such as lack of temperature and dynamics behaviors, hydrogen bonding within a protein-drug complex due to the water and ionic strength of the environment [19,20,36]. MD simulation with full surroundings would provide more realistic factors, and thus the justification of binding behaviors of the compounds towards HER2 could be more consolidated. According to MD simulations, structural analysis and relative binding energy evaluation permitted the following prediction in case trans-(−)-kusunokinin and trans-(+)-kusunokinin could form direct binding with HER2: (1) HER2 preferred trans-(−)-kusunokinin to trans-(+)-kusunokinin, (2) both trans-(−)-kusunokinin and trans-(+)-kusunokinin may bind HER2 differently than HER2 known inhibitors, such as neratinib or synthetic 03q, and (3) trans-(−)-kusunokinin and trans-(+)-kusunokinin exhibited significantly lower binding affinity towards HER2, considering known inhibitors such as neratinib or synthetic 03q.

The well-known HER2 inhibitor binding domain was the so-called ATP binding domain [23,28], consisting of Threonin733 to Leucine785 [37], equivalent to Threonine60 to Leucine112 in our study. Both trans-(−)-kusunokinin and trans-(+)-kusunokinin partly resided in this HER2 domain, unlike 03q and neratinib. The distinctive binding feature of synthetic HER2 inhibitors resulted from several hydrogen bonds in the domain and led to a lower binding energy than kusunokinin. Meanwhile, the trans-(−)-kusunokinin was able to apply a hydrogen bond to Lysine48 in the HER2 structure, whereas no hydrogen bond was observed from trans-(+)-kusunokinin. The reason that only the trans-(−)-form exhibited hydrogen bond formation characteristics was the different spatial orientation of both the trans-(+)-kusunokinin and trans-(−)-kusunokinin in order to fit the HER2 cavity.

The interacting amino acid residues in the ATP binding domain were hydrophobic (nonpolar) amino acids, so the hydrophobic interaction could play a role in ligand selectivity. Moreover, π-π stacking with Phe159 from the aromatic ring in kusunokinin also supported trans-(+)-kusunokinin and trans-(−)-kusunokinin bindings. The π-π stacking with Phe159 was also found in the synthetic inhibitor 03q. The synonymous binding manner of trans-(−)-kusunokinin was also previously documented in the CSF1R [19,20] and AKR1B1 [22]. In short, trans-(−)-kusunokinin and trans-(+)-kusunokinin could bind HER2 in some ATP binding regions using π-π stacking and hydrophobic interactions. Therefore, a reversible HER2 binding model for a case of trans-(−)-kusunokinin was thereby speculated as no covalent bond formation was observed. The previously reported reversible HER2 inhibitors such as lapatinib and tucatinib [38,39,40] can interact with HER2 via hydrogen bonding and π-π stacking, the same as trans-(−)-kusunokinin.

Different scenarios were found in the cases of crotonamide (–C=C–C(=O)–NH–) containing inhibitors, namely neratinib and pyrotinib, which were reported for irreversible HER2 binding [23,40,41,42,43]. The existence of the crotonamide group contributed to the irreversible HER2 binding via covalent bond formation [23,41]. The crotonamide group can react with cysteine100 (Cys100) in the ATP binding domain via Michael addition. There was a carbon-sulfur covalent bond between neratinib and Cys100 in the ATP binding domain (Appendix A), in which the thiol group (-SH) and crotonamide acted as the Michael donor and acceptor, respectively. The mechanism of the reaction is presented in the Appendix A (Appendix A). The irreversible binding of neratinib/pyrotinib could prolong and enhance the action of HER2 inhibition, compared to reversible HER2 binding inhibitor. 

Although an irreversible inhibitor like neratinib would outclass other HER2 inhibitors, since HER2 is overexpressed in 20–30% of primary tumors and associated with malignant transformation and survival rates [44,45], some problems from neratinib were noticed. Neratinib was one of the responsible factors in driving resistance to HER2-targeted therapies [24,25,46], and HER2-positive breast cancer patients usually die due to its resistance [47]. In addition, neratinib toxicity [26,27] is still a main problem. Even though the predicted efficacy of trans-(−)-kusunokinin seemed unable to compete with the irreversible inhibitor (neratinib), the drug-associated drawbacks have left room for further investigation into the anticancer effect of synthetic (±)-kusunokinin based on HER2 and HER2 related proteins in breast cancer. Thus, we decided to carry on the experiment in order to elucidate the anticancer effect of synthetic (±)-kusunokinin by targeting HER2 and its related proteins in similar pathways.

HER2 induces MAPK signaling pathways (RAS-RAF-MEK-ERK) and phosphatidylinositol 3-kinase (PI3K) pathways (PI3K-AKT-mTOR) that result in cell proliferation, angiogenesis and controlled tumor growth [48]. The activation of PI3K-AKT-mTOR, RAS-RAF-MEK-ERK and estradiol (HER2) can enhance the CyclinD-dependent CDK4/6 activity [49]. The activity of CDK4/6 and D-type Cyclins (CyclinD1, CyclinD2 and CyclinD3) drives the cell cycle from G1 to the S phase. The cell cycle progression from G0/G1 to S phase is tightly controlled by c-Myc [50]. CyclinA and CDK1 push cell cycles from the S phase to the G2 phase. Upon mitosis, CDK1 activity is maintained by the complex cyclin B/CyclinB1 [51]. 

Neratinib (tyrosine kinase inhibitor) inhibited the action of proteins by binding with adenosine triphosphate (ATP), leading to the inhibition of phosphorylation on EGFR, HER2, HER3, HER4, AKT and ERK inSKBR3 and BT474 cells (HER2+ breast cancer cells). Moreover, neratinib did not decrease AKT and ERK in SKBR3 cells, meanwhile neratinib completely decreased HER2, AKT, and ERK in BT474 cells [52]. This drug was strongly active against HER-2-overexpressing human breast cancer cells [42]. The Michael acceptor group in neratinib binds EGFR and HER2 at cysteine residues Cys-773 and Cys-805, respectively. Neratinib binds EGFR and HER2 with an irreversible action, as well as afatinib, dacomitinib, and pyrotinib [28,40,43]. 

To confirm the results of our computer simulation, MCF-7 cells were treated with synthetic (±)-kusunokinin and neratinib. The achieved results of this study were consistent with the previous experiment, as described above. The synthetic (±)-kusunokinin showed less toxicity than neratinib on MCF-7 cells. In the silencing of HER2, our study confirms that neratinib strongly bound HER2 by increasing HER2 levels close to the control groups. Moreover, the combination of siRNA-HER2 and neratinib also brought the protein levels of MEK1, ERK, c-Myc, CyclinB1, CyclinD1 and CDK to normal levels. In contrast, the combination of synthetic (±)-kusunokinin and siRNA-HER2 brought only ERK levels to the normal baseline. Regarding biological assays, although trans-kusunokinin was confirmed to have cytotoxic and anti-proliferative activities, it was also shown that kusunokinin suppressed RAS and ERK but not HER2, in contrast with neratinib which is a known ligand of HER2. Therefore, kusunokinin and neratinib are likely to operate via different modes of action, suggesting that kusunokinin may affect HER2-related pathways without physically interacting with the protein, or, in the case that kusunokinin actually binds to HER2, a different binding pocket may be involved, Figure 8.

## 4. Materials and Methods

### 4.1. Compound Acquisition

Trans-(±)-kusunokinin was synthesized following the procedure reported in our previous publication [19]. Neratinib (HKI-272) was obtained from Selleck Chemicals (CAS No: S2150, Houston, TX, USA).

### 4.2. Molecular Docking

Ten proteins associated with breast cancer were chosen for investigating a possible protein target(s) of kusunokinin. The three-dimensional (3D) protein structures were retrieved from the Research Collaboratory for Structural Bioinformatics Protein Database Protein Data Bank (RCSB PDB). All co-crystallized ligands and water molecules were removed using the AutoDock Tool (ADT) version 4.1. All polar hydrogen atoms were added to mimic hydrogen bond interaction, and written into the Protein Data Bank (PDB) file format. 

The 3D structures of trans-(-)-kusunokinin, trans-(+)-kusunokinin and known ligands were taken from the PubChem database. All PubChem CIDs of the ligands in this study were summarized in the Appendix A, Appendix A. All structure files of trans-(−)-kusunokinin, trans-(+)-kusunokinin and known ligands were generated in the PDB file format using the Online SMILES Translator and Structure File Generator (https://cactus.nci.nih.gov/translate/ accessed on 21 January 2021). ADT was used to add all missing polar hydrogen atoms. Finally, both protein and ligand structures were saved in a PDB and Partial Charge (Q) and Atom Type (T), or PDBQT format file.

A molecular docking study between all compounds and the 10 selected proteins was performed using AutoDock4 version 4.2 [53]. The grid box was centered at the region covering the whole structure with the dimensions of 126 × 126 × 126 cubic angstrom (Å^3^). Exhaustiveness was set to 50 docking runs with a population size of 200. All other parameters were set at the default value on the AutoDock4 program. The lowest binding energy (ΔG_bind_) was reported. The lowest energy coordinate was considered for post-docking analysis, and as the starting coordinate of molecular dynamics (MD) simulation.

### 4.3. Molecular Dynamics Simulation

The starting coordinate for MD simulation was adopted from the best docked conformation with the Human HER2 protein database (PDB ID 3PP0) [35] mentioned previously. First of all, the restrained electrostatic atomic partial potential (RESP) charge for all the compounds, such as trans-(+)-kusunokinin, trans-(−)-kusunokinin, synthetic 03q, and neratinib, was modeled based on the optimized geometry and electrostatic single point charges (ESP) using the B3LYP/6-31G* calculation. The B3LYP/6-31G* was implemented in the Gaussian16 package [54], kindly provided by Dr. Thanyada Rungrotemongkol (Department of Biochemistry, Faculty of Science, Chulalongkorn University, Bangkok, Thailand).

Secondly, for both ligand-free HER2 as well as a compound-HER complex structures, all polar hydrogen in the structure from the docking study was removed. All hydrogen atoms were then added again via the Leap module in the AMBER16 package [55]. All ionizable sidechains of amino acids were set as one at pH 7, namely histidine was deprotonated (HIE in AMBER name), lysine and arginine contained + 1e charge, and aspartate and glutamate contained −1e charge. The system was solvated by transferable intermolecular potential with 3 points (TIP3P) water at a distance of 14 angstrom (Å), resulting in approximately 17,400 water molecules and neutralized by either sodium (Na^+^) or chloride (Cl^−^) ion. The 47 NaCl pairs were applied, equivalent to the 0.15 M NaCl solution. 

The MD simulation protocol was carried out similar to the previous kusunokinin study [22]. In brief, the energy minimization was completed using the Steepest Descent method for 1000 steps and the Conjugate Gradient method for 1000 steps under the periodic boundary condition. The canonical (NVT) ensemble at 310 K (37 °C), with a cut off of 16 Å, was used to handle all nonbonded/electrostatic interactions. Harmonic restraint was applied to all coordinates of the compound-protein complex with a force constant of 200, 100, 50, 25 and 10 kcal mol^−1^ Å^−2^, respectively, with reference to the minimized structure. Each force constant lasted for 400 picoseconds (ps) with a time step of 1 femtosecond (fs), resulting in a simulation time of 2 nanoseconds (ns). 

The pressure of 1.013 bar (1 atmospheric pressure) was then introduced to create the isobaric-isothermal (NPT) simulation. The temperature of 310 K and pressure were controlled using the weak-coupling algorithm [56]. All positional restraint was completely removed. The simulation was continued for 100 ns with a time step of 2 fs. The first 50-ns simulation was omitted and the 500 snapshots from the last 50-ns simulation were taken for further structural analysis and binding energy calculation.

### 4.4. Structural Analysis and Free Binding Energy Calculation

The structural analysis was performed via 2 parameters: the root-mean square displacement (RMSD) and the average root-mean square fluctuation (RMSF). The RMSD of the 1000 snapshots from the 100-ns-MD trajectory was illustrated to visualize the simulation stability. In addition, the RMSF of 500 snapshots from the last 50-ns MD trajectory was analyzed to investigate the sidechain flexibility in HER2 proteins, compared with ligand-free states. RMSD was obtained from the Visual Molecular Dynamics (VMD) program [57], while RMSF was calculated from the cpptraj module in the AMBER16 package. 

The free binding energy of all compounds, namely trans-(+)-kusunokinin, trans-(−)-kusunokinin, synthetic 03q, and neratinib, towards HER2 was computed using the molecular mechanics/Poisson-Boltzmann surface area (MM/PBSA) approach [58]. All energetic parameters for the MM/PBSA calculation were summarized in the previous study [59]. The average binding energy (ÄG) was reported in kcal mol^−1^ from the last 500 MD snapshots, using the python script (MMPBSA.py) in the AMBER16 package.

### 4.5. Cell Culture

Breast carcinoma (MCF-7) and normal human epithelial breast (MCF-12A) cell lines were purchased from the American Type Culture Collection (ATCC, Manassas, VA, USA). Mouse fibrosarcoma (L-929) cells were kindly provided by Dr. Jasadee Kaewsrichan, Drug Delivery System Excellence Center, Faculty of Pharmaceutical Sciences, Prince of Songkla University, Songkhla, Thailand. MCF-7 and L-929 cells were cultured in RPMI-1640 and Dulbecco’s modified Eagle medium (DMEM), respectively, and supplemented with 10% fetal bovine serum (FBS), L-glutamine (2 μM), penicillin (100 U/mL) and streptomycin (100 mg/mL) (Invitrogen, Thermo Fisher Scientific, Waltham, MA, USA). MCF-12A cells were maintained in a mixture of DMEM and Ham’s F12 Medium (1:1), supplemented with 5% horse serum, human epidermal growth factor (20 ng/mL), cholera toxin (100 ng/mL), bovine insulin (0.01 mg/mL) and hydrocortisone (500 ng/mL, 95%). All cells were cultured at 37 °C with 5% carbon dioxide in a humidified incubator.

### 4.6. Cytotoxicity Assay

Half of the maximal inhibitory concentration (IC_50_) value was performed as previously described [60]. In brief, cells were seeded in 96 well plates and cultured for 24 h. After treatment, cells were treated at various concentrations of synthetic (±)-kusunokinin and neratinib for 72 h. Cell viability was determined using 3-(4,5-dimethyl thiazol-2 yl)-2,5-diphenyltetrazolium bromide (tetrazolium salt MTT, Cat No.: M6494, Invitrogen, Thermo Fisher Scientific, Waltham, MA, USA). The absorbance was measured at 570 and 650 nM using a microplate reader (Emax^®^ Plus, Molecular Devices, San Jose, CA, USA). Each treatment was performed in triplicate.

### 4.7. In Vitro Transfection of Small Interfering RNA

HER2 gene silencing was performed using siRNA-HER2 (Invitrogen, Thermo Fisher Scientific, Cat No. AM16708A; ID 103546, Carlsbad, CA, USA). The Luciferase duplex (*Photinus pyralis* (American firefly) luciferase gene) (Cat No: P-002099-01-20, Dharmacon, Lafayette, CO, USA) was used as the siRNA control experiment. siRNAs were transfected using Lipofectamine^®^ RNAiMAX (Invitrogen, Waltham, MA, USA), according to the manufacturer’s instructions, as previously described [21]. Briefly, MCF-7 cells were seeded in 24 well plates for 24 h and transfected with 100 nM siRNA-HER2 or 100 nM siRNA-Luciferase. After transfection for 6 h, the medium was removed and changed. Cells were treated with or without IC_50_ concentration values of synthetic (±)-kusunokinin or neratinib for 48 h. Then, effective gene silencing was confirmed by Western blot analysis.

### 4.8. Dye Exclusion Assay

Neratinib (HKI-272) was obtained from Selleck Chemicals (CAS No: S2150, Houston, TX, USA). Cell viability was evaluated by a trypan blue exclusion assay, as previously described [61]. Briefly, MCF-7 cells were treated with synthetic (±)-kusunokinin, neratinib, siRNA-HER2 or a combination of compounds with siRNA-HER2. After the incubation period, floating and attached cells were harvested and stained with trypan blue dye solution at a final concentration of 0.2%. At least 150 cells were counted per treatment using phase-contrast microscopy. Assays were performed in triplicate and repeated 3 times. 

### 4.9. Western Blot Analysis

After treatment, cells were subjected to Western blot analysis, as previously described [61]. Cells were harvested by trypsinization and cells were lysated using a RIPA buffer (Thermo Scientific, Waltham, MA, USA). According to the manufacturer’s instructions, the total protein concentration was determined using the Bradford method (Bio-Rad, Hercules, CA, USA). Eighty micrograms of each protein lysate were loaded and separated on 12% sodium dodecyl sulfate-polyacrylamide gel electrophoresis (SDS-PAGE) and transferred to a nitrocellulose membrane (Millipore, Billerica, MA, USA). To prevent non-specific binding, the membranes were blocked with 5% non-fat dry milk in TBST (0.1% Tween 20, 154 mM NaCl, 48 mM Tris-base, pH 6.8). The membranes were probed with primary antibodies including anti-HER2, c-Myc, CyclinD1, Ras, AKT, MEK1 (Cell Signaling Technology, Danvers, MA, USA), CDK1, CyclinB1, ERK (Santa Cruz Biotechnology, Dallas, TX, USA) and GAPDH antibodies (Calbiochem, Darmstadt, Germany). The protein signal was visualized using the SuperSignal™ West Dura Extended Duration substrate kit (Thermo Scientific, Waltham, MA, USA), according to the protocol supplied with the kit. Subsequently, immunoblot images were quantified using the ImageJ program (NIH Image, Bethesda, MD, USA).

### 4.10. Statistical Analysis

Data values of 3 independent experiments are expressed as the mean ± standard deviation (SD). The student’s *t*-test on Microsoft Excel was used for statistical analysis. A *p*-value of less than 0.05 was considered to indicate a statistically significant difference between groups.

## 5. Conclusions

Trans-(−)-kusunokinin had the potential to interact with the ATP binding domain of HER2 in which nonbonded interactions played an essential part in HER2 binding. Trans-(−)-kusunokinin had better HER binding affinity compared to its isomer trans-(+)-kusunokinin. Computational modeling predicted that trans-(−)-kusunokinin could be active against HER2 as a reversible HER2 inhibitor. The synthetic (±)-kusunokinin exhibited lower toxicity than neratinib on MCF-7, MCF-12A and L-929 cells. This compound suppressed RAS, ERK, CyclinB1, CyclinD1 and CDK1. In silencing HER2, synthetic (±)-kusunokinin suppressed c-Myc, CyclinB1, CyclinD1 and CDK1. Meanwhile, neratinib alone decreased HER2, MEK1, ERK, c-Myc CyclinB1, CyclinD1 and CDK1. Moreover, neratinib did not suppress all the tested proteins during HER2 silencing. In summary, synthetic (±)-kusunokinin exhibited low HER2 binding affinity with a different mode of action from HER2 inhibitor, neratinib.

## Figures and Tables

**Figure 1 molecules-26-04537-f001:**
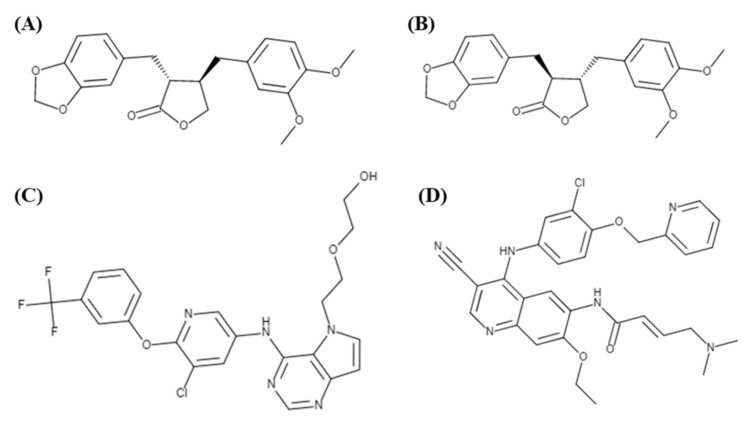
Ligands used in HER2 MD simulations: (**A**) trans-(−)-kusunokinin, (**B)** trans-(+)-kusunokinin, (**C)** synthetic inhibitor 03q, and (**D**) neratinib (HKI-272).

**Figure 2 molecules-26-04537-f002:**
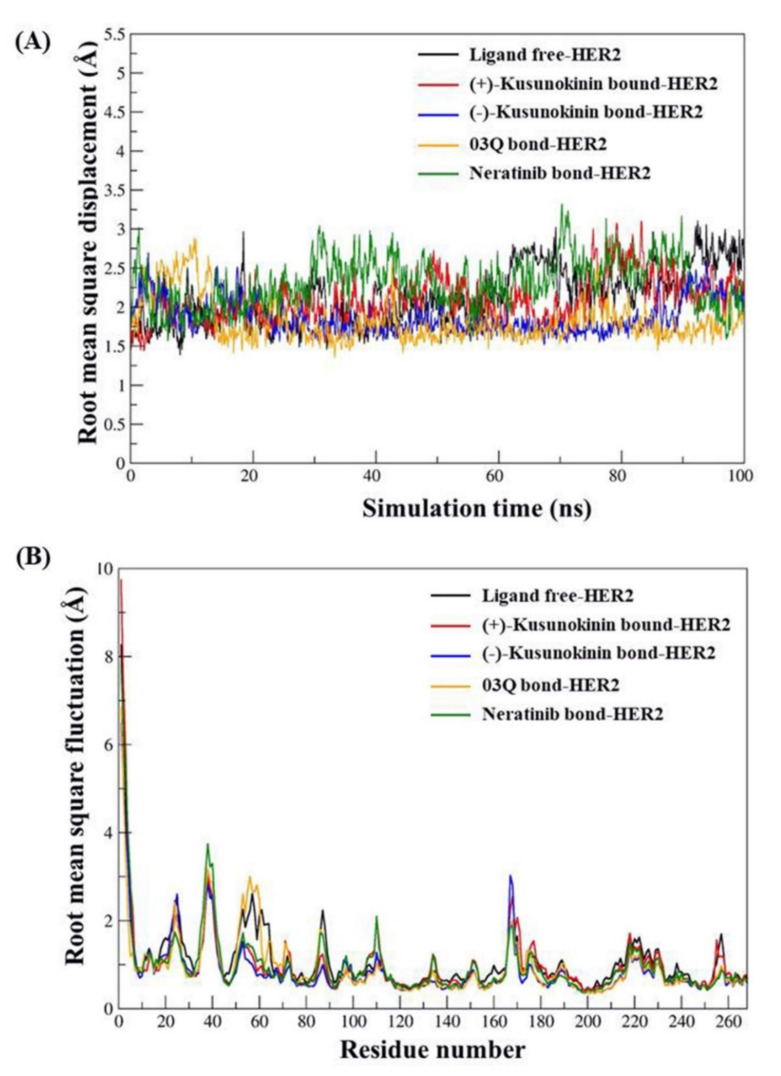
Root mean square displacement (RMSD) and root mean square fluctuation (RMSF) from ligand bound HER2 simulations. (**A**) RMSD plot was computed from 100 ns MD trajectories. (**B**) RMSF plot was derived from an alpha carbon atom of each amino acid in the HER2 structure. Both RMSD and RMSF were reported in a units of angstrom (Å).

**Figure 3 molecules-26-04537-f003:**
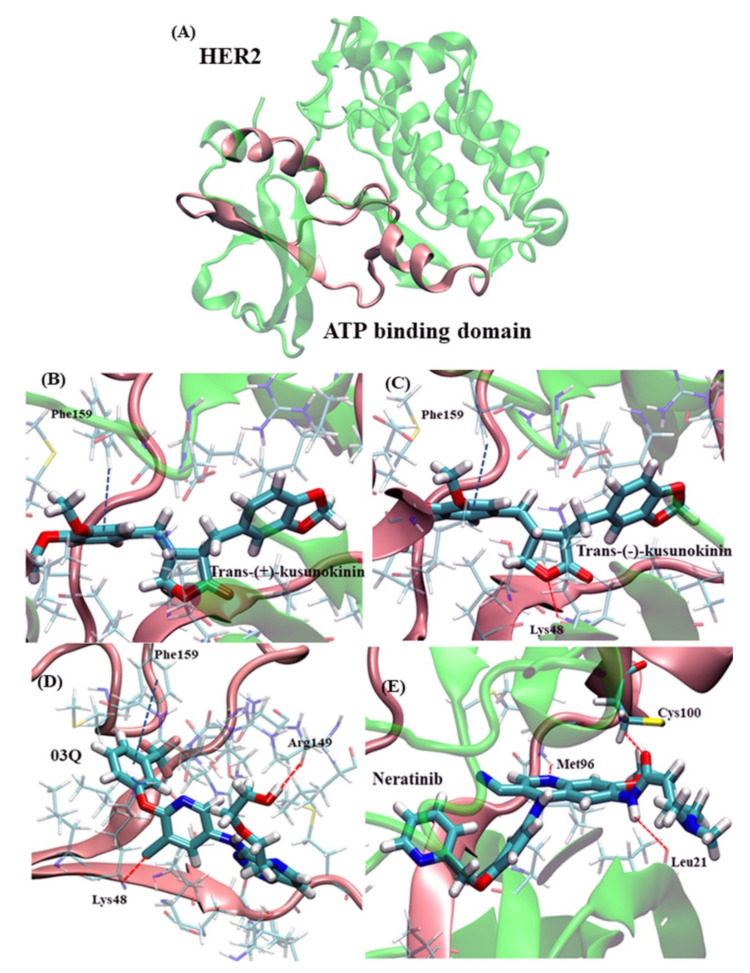
Ligand interaction with HER2 structure. (**A**) HER2 structure and its ATP binding domain (pink), (**B**) trans-(+)-kusunokinin, (**C**) trans-(−)-kusunokinin, (**D**) 03q and (**E**) neratinib. The red line represents hydrogen bonding or hydrogen bond like interaction (red line). Phe159 was responsible for π-π interaction (blue line). The transparent amino acid represents amino acids interacting with the compound through hydrophobic interaction.

**Figure 4 molecules-26-04537-f004:**
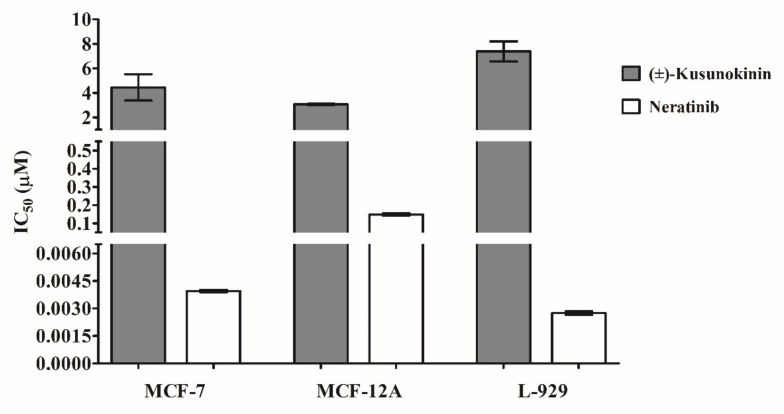
Cytotoxicity effect of synthetic (±)-kusunokinin and neratinib. Three different cell lines (MCF-7, MCF-12A and L-929) were incubated with tested compounds and incubated for 72 h. The cytotoxicity effects of each compound were determined using an MTT assay. Data represent the mean ± SD of three independent experiments.

**Figure 5 molecules-26-04537-f005:**
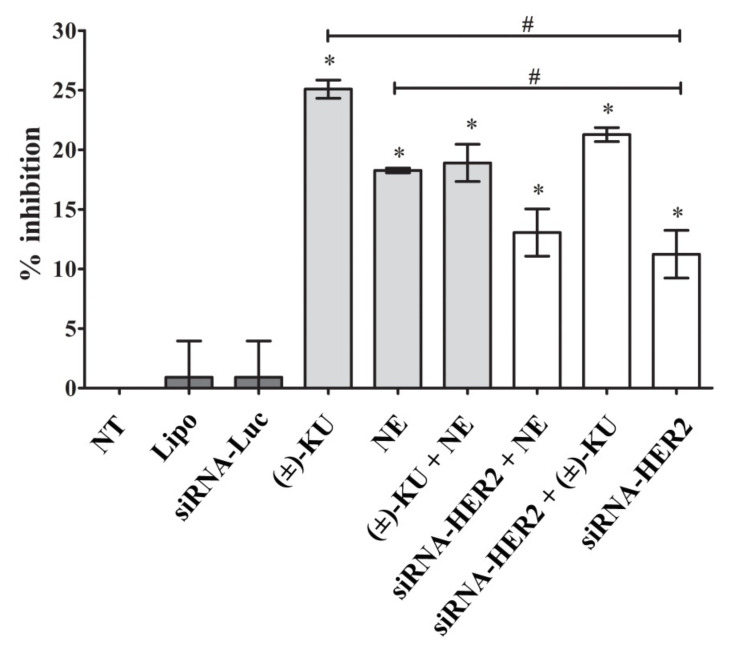
The inhibitory effect of the compounds, siRNA-HER2 and the combination on breast cancer cells. MCF-7 cells were seeded in 48 well plates with 3 × 10^4^ cells/well and treated with IC_50_ values of tested compounds (4.45 µMsynthetic (±)-kusunokinin, 0.004 µMneratinib, or 4.45 µM (±)-kusunokinin combination with 0.004 µMneratinib),100 nM siRNA-HER2, and 100 nM siRNA-HER2 combination with IC_50_ values of (±)-kusunokinin, or neratinib. Cells treated with transfection reagent (lipofectamine), siRNA-Luciferase or non-treated were the control group. Cell viability was investigated using the trypan blue exclusion assay. The percentage of cell inhibition represents the mean ± SD of three independent experiments. The student’s *t*-test was used for the consideration of *p* values. * *p* < 0.05 vs. control and ^#^
*p* < 0.05 vs. siRNA-HER2. NT, non-treated cells; Lipo, lipofextamine; siRNA-Luc, siRNA-Luciferase; (±)-KU, (±)-kusunokinin; NE, neratinib; (±)-+NE, (±)-kusunokinin+neratinib; siRNA-HER2+NE, siRNA-HER2+neratinib; siRNA-HER2 + (±)-KU, siRNA-HER2 + (±)-kusunokinin.

**Figure 6 molecules-26-04537-f006:**
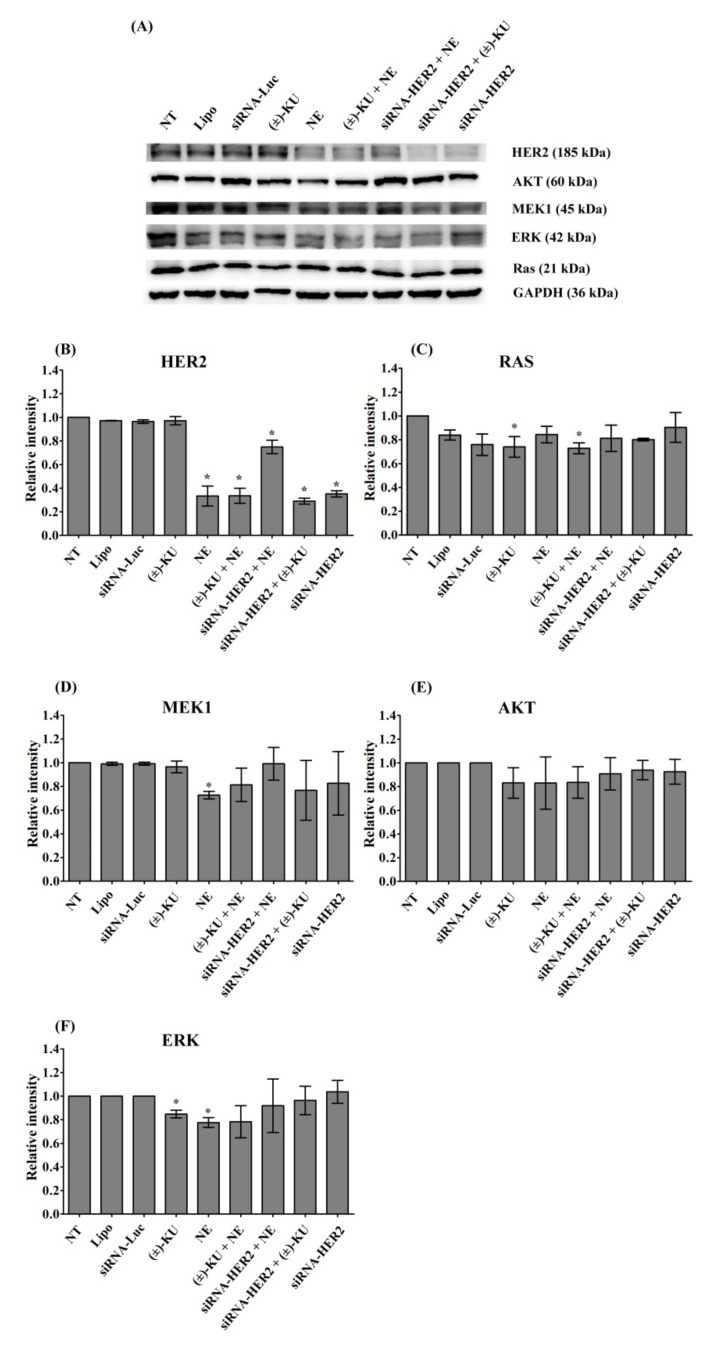
Effect of synthetic (±)-kusunokinin on HER2 and its downstream proteins. MCF-7 cells were treated with synthetic (±)-kusunokinin (4.45 μM), neratinib (0.004µM), a combination of synthetic (±)-kusunokinin and neratinib, or100 nM siRNA-HER2 for 48 h. For the combination of siRNA the with compound, cells were incubated with 100 nM of siRNA-HER2 for 24 h followed by treatment with 4.45 μM (±)-kusunokinin, or 0.004 µM neratinib for 48 h. The negative controls were non-treated cells, lipofectamine and siRNA-luciferase treatments. (**A**) After incubation, cells were harvested and lysed by RIPA buffer. Total protein was subjected to SDS-PAGE and proteins of interest were determined using Western blot analysis. GAPDH served as an internal control. The quantitative proteins of interest, including (**B**) HER2, (**C**) Ras (**D**) MEK1, (**E**) AKT, and (**F**) ERK, were calculated by normalized with GAPDH band intensity. Data are representative of three independent experiments. All graphs show mean ± S.D. The student’s *t*-test was used for consideration of *p* values < 0.05. * *p* < 0.05 vs. control. NT, non-treated cells; Lipo, lipofextamine; siRNA-Luc, siRNA-Luciferase; (±)-KU, (±)-kusunokinin; NE, neratinib; (±)-+NE, +(±)-kusunokinin+neratinib; siRNA-HER2+NE, siRNA-HER2+neratinib; siRNA-HER2 + (±)-KU, siRNA-HER2 + (±)-kusunokinin.

**Figure 7 molecules-26-04537-f007:**
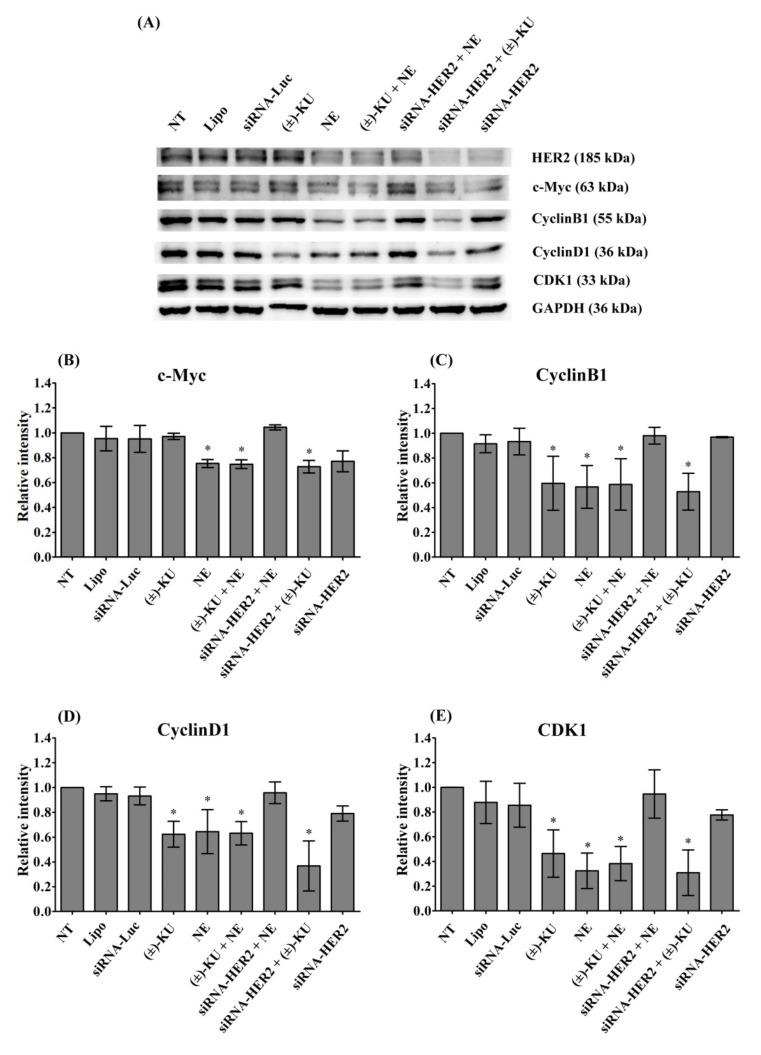
Effect of synthetic (±)-kusunokinin on cell proliferation proteins. MCF-7 cells were treated with synthetic (±)-kusunokinin (4.45 μM), neratinib (0.004µM), a combination of synthetic (±)-kusunokinin and neratinib, or 100 nM siRNA-HER2 for 48 h. For the combination of siRNA with compound, cells were incubated with 100 nM of siRNA-HER2 for 24 h followed by treatment with 4.45 μM (±)-kusunokinin, or 0.004 µM neratinib for 48 h. The negative controls were non-treated cells, lipofectamine and siRNA-luciferase treatments. (**A**) After incubation, cells were harvested and lysed by RIPA buffer. Total protein was subjected to SDS-PAGE and proteins of interest were determined using Western blot analysis. GAPDH served as an internal control. The quantitative proteins of interest, including (**B**) c-Myc, (**C**) CyclinB1, (**D**) CyclinD1, and (**E**) CDK, were calculated by normalized with GAPDH band intensity. Data are representative of three independent experiments. All graphs show mean ± S.D. The student’s *t*-test was used for consideration of *p* values < 0.05. * *p* < 0.05 vs control. NT, non-treated cells; Lipo, lipofextamine; siRNA-Luc, siRNA-Luciferase; KU, (±)-kusunokinin; NE, neratinib; KU+NE, (±)-kusunokinin+neratinib; siRNA-HER2+NE, siRNA-HER2+neratinib; siRNA-HER2+KU, siRNA-HER2+(±)-kusunokinin.

**Figure 8 molecules-26-04537-f008:**
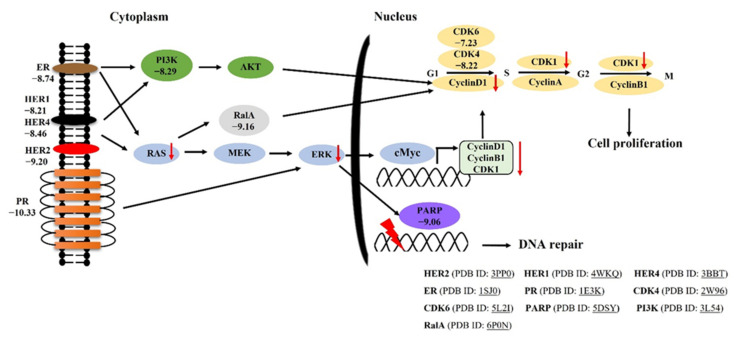
The proposed pathway of trans-(−)-kusunokinin binds HER2 protein on breast cancer cells. A number denotes the binding energy in kcal/mol from molecular docking analysis. The red arrow refers to the decreasing of proteins on MCF-7 cells by trans-(−)-kusunokinin. The PDB identification code of each protein is shown as the underlined text.

**Table 1 molecules-26-04537-t001:** Docking score of trans-(±)-kusunokinin and inhibitor with 10 breast cancer-associated proteins.

Target Protein	Known Inhibitor	Docking Score (kcal/mol)		
		Inhibitor	(−)-Kus ^1^	(+)-Kus ^2^
Human epidermal growth factor receptor 2 (HER2)	03Q Neratinib	−9.13 −8.88	−9.20 −9.20	−9.16 −9.16
Human epidermal growth factor receptor 1 (HER1)	Gefitinib	−7.69	−8.21	−8.03
Human epidermal growth factor receptor 4 (HER4)	Lapatinib	−10.37	−8.46	−8.64
Estrogen receptor (ER)	E4D	−13.55	−8.74	−8.87
Progesterone receptor (PR)	R18	−11.25	−10.33	−10.02
Cyclin-dependent kinases 4 (CDK4)	Palbociclib	−9.02	−8.22	−9.09
Cyclin-dependent kinases 6 (CDK6)	LQQ	−9.12	−7.23	−8.31
Poly (ADP-ribose) polymerase (PARP)	UHB	−11.27	−9.06	−9.13
Phosphoinositide−3 kinase (PI3K)	LXX	−8.46	−8.29	−9.15
Ras-related protein Ral-A (RalA)	NLS	−9.64	−9.16	−9.33

^1^ trans-(−)-kusunokinin. ^2^ trans-(+)-kusunokinin.

**Table 2 molecules-26-04537-t002:** Binding energies of ligand-HER complexes obtained from molecular dynamics simulations.

Inhibitor	MM/GBSA(kcal/mol)	MM/PBSA(kcal/mol)
Trans-(+)-kusunokinin	−39.03 ± 0.14	−29.18 ± 0.18
Trans-(−)-Kusunokinin	−44.06 ± 0.14	−29.79 ± 0.15
03Q	−54.28 ± 0.17	−40.78 ± 0.22
Neratinib	−50.44 ± 0.18	−40.75 ± 0.21

## Data Availability

Not applicable.

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
