# Peer review of "Trans-(−)-Kusunokinin: A Potential Anticancer Lignan Compound against HER2 in Breast Cancer Cell Lines?"

_molecules, 2021, doi:10.3390/molecules26154537_

Round 1
Reviewer 1 Report
The present paper focuses on trans-kusunokinin, an anticancer lignan compound of importance in breast cancer, whose protein target has not yet been clearly determined. To address this question, which I think is of significance, the authors have conducted a study involving cytotoxicity and proliferation assays, western blot and computational modeling. Although the study is clear and well presented, I do not think the presented results are sufficient to validate the main conclusion of the paper, namely, that kusunokinin’s mode of action is done through direct (i.e., physical) binding to HER2. First, computational methods such as docking and PBSA/GBSA calculations, are usually not intended to validate whether a molecular compound corresponds to a protein ligand but rather will provide estimates of the binding mode and (relative) binding energy of an already-known binder (although, e.g., in virtual screening scenarios, we expect top hits to be actual ligands for the target of interest). In my opinion, the result obtained by the authors regarding docking scores being lower (i.e., better) than 03Q and Neratibin in the case of HER2 (Table 1) is not significative of the ability of kusunokinin to bind to HER2 as docking scores are normally used to select the best pose from one docking run and do not necessarily correlate with the binding affinity (for this reason, it is, in my opinion, misleading to interpret docking scores as binding energies as the authors did in line 80). This argument is also confirmed by GBSA/PBSA calculations (such methods usually provide a better correlation with pKd values compared to docking) where the authors found that the predicted binding energies were much weaker than those of 03Q and Neratibin (Table 2). Again, GBSA/PBSA scores are not sufficient to confirm the binding ability of trans-kusunokinin to HER2. Regarding biological assays, although trans-kusunokinin was confirmed by the authors to have a cytotoxic and anti-proliferative activity, it was also shown that kusunokinin is suppressing RAS and ERK but not HER2 in contrast with neratinib which is a known ligand of HER2. Therefore, kusunokinin and neratinib are likely to operate via different modes of action suggesting that kusunokinin may affect HER2-related pathway without physically interacting with the protein, or, in case kusunokinin actually binds to HER2, a different binding pocket may be involved. In their study, it looks like the authors performed docking on a single protein structure, i.e., the flexibility of the binding pockets is not taken into consideration, which might prevent docking in other areas. A more rigorous approach would consist in running molecular dynamics (MD) simulation prior to docking in order to generate representative structures of the protein which account for the flexibility of the target.
Other remark: in my opinion, English language should be improved (the manuscript should be proofread by a native English speaker). There are also many typos and phrases that are not understandable. Examples: Line 26: the mechanism of (±)-kusunokinin partially like neratinib. Line 135: as the same site as. Line 153: the MM/PBSA and MM/GBSA approach had predicted that trans(-)-kusunokinin could be an active form towards HER2. Also, it is not clear what the authors mean by “partial binding”, does it mean a ligand with low affinity?
Author Response
Comments and Suggestions for Authors
Q1. The present paper focuses on trans-kusunokinin, an anticancer lignan compound of importance in breast cancer, whose protein target has not yet been clearly determined. To address this question, which I think is of significance, the authors have conducted a study involving cytotoxicity and proliferation assays, western blot and computational modeling. Although the study is clear and well presented, I do not think the presented results are sufficient to validate the main conclusion of the paper, namely, that kusunokinin’s mode of action is done through direct (i.e., physical) binding to HER2.
Response: Thanks for the suggestions. We changed the conclusion of the study to be more reasonable and consistent with the newly correct interpretation. The conclusion was also rewritten to be corresponding to the results in the manuscript. The two sentences, which are “Trans-(-)-kusunokinin had the potential to interact with ATP binding domain of HER2 in which nonbonded interactions played an essential part in HER2 binding (page 22, line 650-652).” and “In summary, synthetic (±)-kusunokinin exhibited low HER2 binding affinity with a different mode of action from HER2 inhibitor, neratinib (page 22, line 662-664).”, could support the results more and lead to the less overclaimed statement.
We changed the title to “Trans-(+)-kusunokinin: An anticancer lignan against HER2 in breast cancer cell lines?” so that it will not be made the reader misunderstand our concluding remarks.
Q2. First, computational methods such as docking and PBSA/GBSA calculations, are usually not intended to validate whether a molecular compound corresponds to a protein ligand but rather will provide estimates of the binding mode and (relative) binding energy of an already-known binder (although, e.g., in virtual screening scenarios, we expect top hits to be actual ligands for the target of interest).
Q2.1 In my opinion, the result obtained by the authors regarding docking scores being lower (i.e., better) than 03Q and Neratib in the case of HER2 (Table 1) is not significant of the ability of kusunokinin to bind to HER2 as docking scores are normally used to select the best pose from one docking run and do not necessarily correlate with the binding affinity (for this reason, it is, in my opinion, misleading to interpret docking scores as binding energies as the authors did in line 80).
Response: Thanks for the suggestion. We rewrote this part to correct the interpretation misleading. The docking score was based on the “relative binding energy” instead of “binding energy”. The words “stronger” and “lower binding energy” were removed to avoid misunderstanding. The sentence in the new manuscript was “The docking score was represented by the relative binding energy (ΔGbind) in kcal/mol. HER2 became of interest as the first prioritized target as trans-(±)-kusunokinin because HER2 were suggested as the challenging targets for breast cancer therapy [23] due to its drug resistance, cross-resistance to other HER2-targeted drugs [24,25] and drug toxicity [26,27].” (Page 3, lines 99-104).
We also wrote that molecular docking only guides the potential binding site, and the evaluation of the binding behavior was examined from MD simulation. This mentioned sentence was “Molecular docking could, however, only roughly guide the potential binding site of the drug towards an interested protein. MD simulation was thus undertaken in order to circumvent the prediction of the relative binding energy of the compounds compared to the known inhibitors, and the inspection of binding behaviors under a more realistic condition.” (Page 3, lines 108-113).
Q2.2 This argument is also confirmed by GBSA/PBSA calculations (such methods usually provide a better correlation with pKd values compared to docking) where the authors found that the predicted binding energies were much weaker than those of 03Q and Neratib in (Table 2). Again, GBSA/PBSA scores are not sufficient to confirm the binding ability of trans-kusunokinin to HER2.
Response: Thanks for the key suggestion. We have changed the interpretation of binding justification into the binding tendency. We have applied GBSA/PBSA to evaluate the possibility of HER2 binding of synthetic (±)-kusunokinin and changed the “binding energy” into “relative binding energy”. We have also emphasized the correct interpretation of the relative binding energy on page 6 lines 175-185 at the sentence “The relative binding energy was then evaluated from MM/GBSA and MM/PBSA based on the MD trajectory. To be noted, the relative binding energy was just able to predict the relative binding tendency of the ligand to the target protein, not to identify whether the compound can truly bind the studied target. Rather, in our study, we could predict the relative possibility of synthetic (±)-kusunokinin for HER2 binding with respect to the known inhibitors. The relative HER2 binding energies of trans-(-)-kusunokinin and trans-(+)-kusunokinin were then acquired, whereas neratinib and synthetic 03q were taken as the reference binding energy (Table 2).”.
Q3. Regarding biological assays, although trans-kusunokinin was confirmed by the authors to have a cytotoxic and anti-proliferative activity, it was also shown that kusunokinin is suppressing RAS and ERK but not HER2 in contrast with neratinib which is a known ligand of HER2. Therefore, kusunokinin and neratinib are likely to operate via different modes of action suggesting that kusunokinin may affect HER2-related pathway without physically interacting with the protein, or, in case kusunokinin actually binds to HER2, a different binding pocket may be involved.
Response: Thank you for your suggestion. We agreed with the reviewer and we added this suggestion in “Discussion” as following “According to MD simulations, structural analysis and relative binding energy evaluation permitted the following prediction in case trans-(-)-kusunokinin and trans-(+)-kusunokinin could form direct binding with HER2: 1) HER2 preferred trans-(-)-kusunokinin to trans-(+)-kusunokinin, 2) Both trans-(-)-kusunokinin and trans-(+)-kusunokinin may bind HER2 differently than HER2 known inhibitors, such as neratinib or synthetic 03q, and 3) trans-(-)-kusunokinin and trans-(+)-kusunokinin exhibited significantly lower binding affinity towards HER2, considering known inhibitors such as neratinib or synthetic 03q.” (Page 15, lines 355-364).
One more paragraph was added for the last paragraph of the “Discussion” as “Regarding biological assays, although trans-kusunokinin was confirmed to have cytotoxic and anti-proliferative activities, it was also shown that kusunokinin suppressed RAS and ERK but not HER2, in contrast with neratinib which is a known ligand of HER2. Therefore, kusunokinin and neratinib are likely to operate via different modes of action, suggesting that kusunokinin may affect HER2-related pathways without physically interacting with the protein, or, in the case that kusunokinin actually binds to HER2, a different binding pocket may be involved, Figure 8.” to emphasize reviewer suggestion (Page 17, lines 459-467).
Q4. In their study, it looks like the authors performed docking on a single protein structure, i.e., the flexibility of the binding pockets is not taken into consideration, which might prevent docking in other areas. A more rigorous approach would consist in running molecular dynamics (MD) simulation prior to docking in order to generate representative structures of the protein which account for the flexibility of the target.
Response: We understand this point of view. Two perspectives were reported in our work. In the first of this study, we performed blind docking along with the crystal structure as a rigid receptor while the ligand was flexible. We decided to use molecular docking to predict the possible pose from 5 GA runs of which each run contained 10000 independent samplings. We assumed that 50,000 poses would be enough for obtaining the possible pose for the drug-HER2 complex. After that, we investigated dynamics behaviors from MD simulation as we did in this study and computed relative binding energy.
Later, we experimented as the reviewer suggested. We extracted 10 equidistant snapshots from the last 50 ns MD trajectory of ligand-free HER2. We did molecular docking as in the study on the chosen MD snapshots. We found that most of the best poses in each MD taken snapshot of trans-(-)-kusunokinin went into a similar pocket as previous molecular docking predicted. This can be implied that the molecular docking of the ligand on the rigid HER2 followed by the MD simulation showed would be sufficient to identify the possible binding site, at least in this case. According to these mentioned results, we supposed that the binding site of kusunokinin was as reported in this study, and was different from the HER2 inhibitors.
Q5. Other remark: in my opinion, English language should be improved (the manuscript should be proofread by a native English speaker). There are also many typos and phrases that are not understandable.
Response: We have rewritten some contents in our manuscript. Our manuscript has corrected the grammar and writing by native English staff at the Department of International Affairs, Faculty of Medicine, Prince of Songkla University.
Q5.1 Examples: Line 26: the mechanism of (±)-kusunokinin partially like neratinib.
Response: We changed the sentence on page 1 lines 33-34 “the mechanism of (±)-kusunokinin partially like neratinib.” into “We conclude that (±)-kusunokinin bound HER2 with low affinity and had a different action from neratinib”.
Q5.2 Line 135: as the same site as
Response: We changed the sentence on page 5 (lines 159-161) “Trans-(-)-kusunokinin and trans-(+)-kusunokinin had bound the same HER2 site, using hydrophobic interaction and π-π stacking due to Phenylalaine159 (Phe159).”
Q5.3 Line 153: the MM/PBSA and MM/GBSA approach had predicted that trans-(-)-kusunokinin could be an active form towards HER2.
Response: This sentence was removed from the manuscript.
Q5.4 Also, it is not clear what the authors mean by “partial binding”, does it mean a ligand with low affinity?
Response: We have replaced “partial binding” with “low affinity” in the "Abstract" (page 1, line 33) and "Conclusion" (page 22, line 666).

Reviewer 2 Report
In this paper represents authors combine two different approaches (bioinformatics and in-vitro tests) to provide information on the kusunokokinin possible targets.
However, the manuscript suffers from some inconsistency, which makes it difficult to identify the main message.
A carefully re-editing of the manuscript will further improve the quality of the study
Minor points
Clarify the sentence "kusunokinin partially bound HER2"
Clarify the sentence "The levels of HER2 and 5 proteins "
Some parts of figure 10 are not visible
PDB IDs are missing
Please further elucidate the sentence
“In summary, synthetic (±)-kusunokinin was not directly bound to HER2 and had different actions with neratinib. Further studies will be investigated the direct binding of trans-(-) and trans-(+)-kusunokinin with HER2 compared with neratinib”
Author Response
Comments and Suggestions for Authors
Q1. In this paper represents authors combine two different approaches (bioinformatics and in-vitro tests) to provide information on the kusunokinin possible targets. However, the manuscript suffers from some inconsistency, which makes it difficult to identify the main message. A carefully re-editing of the manuscript will further improve the quality of the study.
Response: Thank you for the suggestion. We have rewritten some contents in our manuscript. Our manuscript has corrected the grammar and writing by native English staff at the Department of International Affairs, Faculty of Medicine, Prince of Songkla University.
Minor points
M-Q1. Clarify the sentence "kusunokinin partially bound HER2"
Response: We have replaced “kusunokinin partially bound HER2” with “low affinity” in the "Abstract" (page 1, line 33) and "Conclusion" (page 22, line 666).
M-Q2. Clarify the sentence "The levels of HER2 and 5 proteins"
Response: We changed the sentence "The levels of HER2 and 5 proteins" to “The protein levels of HER2, Ras, MEK1, AKT and ERK were measured by Western blot analysis (Figure 6A).” (Page 11, lines 263-264).
M-Q3. Some parts of figure 10 are not visible, PDB IDs are missing.
Response: We have corrected Figure 10 along with PDB identification codes. Figure 10 was newly changed into Figure 8 in the revised manuscript (Page 18).
M-Q4. Please further elucidate the sentence “In summary, synthetic (±)-kusunokinin was not directly bound to HER2 and had different actions with neratinib. Further studies will be investigated the direct binding of trans-(-) and trans-(+)-kusunokinin with HER2 compared with neratinib”
Response: We have changed the sentence in the conclusion section into “In summary, synthetic (±)-kusunokinin was bound HER2 with low affinity and had different actions with neratinib.” into “In summary, synthetic (±)-kusunokinin exhibited low HER2 binding affinity with a different mode of action from HER2 inhibitor, neratinib.” (Page 22, lines 666-668) to avoid misunderstanding.

Reviewer 3 Report
This paper presents information about the effects of trans-(-)-kusunokinin as an anticancer chemical. Although it contains good data, it requires major English Editing that should be carried out before acceptance.
Things such as "investigating the target protein of trans-(-)-kusunokinin is still worth trying" should be avoided in the Abstract.
The ligands (Column 3) in Table 1 should be defined. Use Suppl. Material.
Table 1. Please change "Receptor of Breast cancer"
This part doesn´t have support: "We conclude that (±)-kusunokinin partially bound HER2." Please re-write it.
Remove background in lines 70-75.
Figures 8 and 9 are basically speculations. These have no support on the paper. I suggest to send them to Suppl. Material.
In the Conclusion, line 516. I would prefer not to state that "Trans-(-)-kusunokinin interacted with ATP binding domain of HER2". The data is just in silico. This is not defintive proof, and it is a major bias in the paper. The authors should use the words "Trans-(-)-kusunokinin has the potential to interact with...".
Author Response
Comments and Suggestions for Authors
Q1. This paper presents information about the effects of trans-(-)-kusunokinin as an anticancer chemical. Although it contains good data, it requires major English Editing that should be carried out before acceptance.
Response: Our manuscript has corrected the grammar and writing by native English staff at the Department of International Affairs, Faculty of Medicine, Prince of Songkla University.
Q2. Things such as "investigating the target protein of trans-(-)-kusunokinin is still worth trying"
should be avoided in the Abstract.
Response: We changed the sentence “investigating the target protein of trans-(-)-kusunokinin is still worth trying” into “Therefore, finding an additional possible target of trans-(-)-kusunokinin remains of importance for further development.” (Page 1, lines 13-14)
Q3. The ligands (Column 3) in Table 1 should be defined. Use Suppl. Material.
Response: We changed the word “ligand” into “known inhibitor”. We also put the PubChem CID for these compounds in the supplementary material (Table S1).
“All PubChem CIDs of the ligands in this study were summarized in supplementary material, Table S1.” (Page 18, line 487-489)
Q4. Table 1. Please change "Receptor of Breast cancer"
Response: We removed "Receptor of Breast cancer" to avoid reader confusion. We put only the protein names instead. (Table 1 on page 4)
Q5. This part doesn´t have support: "We conclude that (±)-kusunokinin partially bound HER2." Please re-write it.
Response: We have rewritten "We conclude that (±)-kusunokinin partially bound HER2." into “We conclude that (±)-kusunokinin bound HER2 with low affinity and had the different action from neratinib.”. (Page 1, lines 34-33)
Q6. Remove background in lines 70-75.
Response: We removed these sentences in lines 70-75 and replaced one sentence at the end of the introduction to inform an aim of the study as “In this recent study, we performed a computer simulation and an in vitro study in order to determine the target protein of racemic trans-kusunokinin in breast cancer.” (page 3, lines 88-91).
The background in lines 70-75 was added into the first part of “Discussion” to inform some ideas of the study “In our previous work, trans-(-)-kusunokokinin was a potential compound for cancer treatment. However, many points remain unclear, especially verified target proteins. To develop this compound to be a targeted drug for breast cancer, specific target identification was necessary for 3 reasons: 1) breast cancer comprises many receptors that relate to cancer treatment, 2) (-)-kusunokinin did not show strong binding with CSF1R and AKR1B1, and 3) synthetic (±)-kusunokinin contained 2 forms. Therefore, finding other target proteins from breast cancer associated proteins was the main objective of this work.” (Page 12 Line 300-309).
Q7. Figures 8 and 9 are basically speculations. These have no support on the paper. I suggest to send them to Suppl. Material.
Response: We removed Figure 8 because the written texts were enough to give the necessary information. Figure 9 was sent to supplementary material as a Figure S1.
Q8. In the Conclusion, line 516. I would prefer not to state that "Trans-(-)-kusunokinin interacted with ATP binding domain of HER2". The data is just in silico. This is not definitive proof, and it is a major bias in the paper. The authors should use the words "Trans-(-)-kusunokinin has the potential to interact with...".
Response: We changed the sentence “Trans-(-)-kusunokinin interacted with ATP binding domain of HER2 in which nonbonded interactions played an essential part in HER2 binding.” to “Trans-(-)-kusunokinin had the potential to interact with the ATP binding domain of HER2 in which nonbonded interactions played an essential part in HER2 binding. Trans-(-)-kusunokinin had better HER binding affinity compared to its isomer trans-(+)-kusunokinin. Computational modeling predicted that trans-(-)-kusunokinin could be active against HER2 as a reversible HER2 inhibitor”. (Page 22, lines 654-659).

Round 2
Reviewer 1 Report
The authors have addressed my main concerns. However, there are still a couple of stylistic issues that should be resolved before possible acceptance:
Please use present tense to describe biological/computational background, and past tense to describe your results. Alternatively, you can use “was shown/observed/reported to” when talking about someone else’s results. For example, from line 71: Trans-(-)-kusunokinin, a lignan compound, isolated from Piper nigrum inhibits breast cancer cell growth, and induces cell cycle blockage, apoptosis and cell cycle arrest at the Gap 2/mitotic (G2/M) phase [18]. the synthetic compound of trans-kusunokinin consists of 2 forms, including trans-(-)-kusunokinin and trans-(+)-kusunokinin. This synthetic compound exhibits similar cytotoxic activity on breast and colon cancer, same as the extract. The synthetic trans-(±)- kusunokinin was found to suppress topoisomerase II, STAT3, CyclinD1 and p21 on breast cancer and cholangiocarcinoma cells”. Please correct this issue in the other paragraphs as well including the abstract. For example, line 1 “Trans-(-)-kusunokinin, an anticancer compound, partially binds CSF1R in breast cancer cells”. Regarding this last sentence, please remove “partially bind” everywhere in the text as it is highly misleading for the reader. A compound binds to a target (possibly with low affinity) or not.
Other phrases should be rewritten as they remain unclear:
Line 99: replace “The docking score was represented by the relative binding energy (ΔGbind) in kcal/mol” by “The docking score was given as an effective binding energy (ΔGbind) in kcal/mol although it may differ from the true binding energy”.
Line 111: “in order to circumvent the prediction of the relative binding energy”. The verb circumvent normally applies to a rule or a problem not a prediction. Please correct.
Line 113: “under more realistic conditions”
Paragraph from line 130: replace MD simulations by MD trajectories.
Line 141: “The RMSD plot denoted the stability of MD simulation after 50 ns of simulation time since the RMSD values in all 5 simulations constantly fluctuated (Figure 2A). Hence, taking the last 50 ns simulation was valid in terms of both structure and binding analysis.” Stability of a system is normally verified by checking that the RMSD plateaus soon after the beginning of the simulation with relatively low RMSD value (<3.0Å or so). Is it the case here? Also, it is not clear why Figure 2A does not start from 0 RMSD.
Author Response
Reviewer #1
Comments and Suggestions for Authors
Q1. The authors have addressed my main concerns. However, there are still a couple of stylistic issues that should be resolved before possible acceptance:
Please use present tense to describe biological/computational background, and past tense to describe your results. Alternatively, you can use “was shown/observed/reported to” when talking about someone else’s results. For example, from line 71: Trans-(-)-kusunokinin, a lignan compound, isolated from Piper nigrum inhibits breast cancer cell growth, and induces cell cycle blockage, apoptosis and cell cycle arrest at the Gap 2/mitotic (G2/M) phase [18]. The synthetic compound of trans-kusunokinin consists of 2 forms, including trans-(-)-kusunokinin and trans-(+)-kusunokinin. This synthetic compound exhibits similar cytotoxic activity on breast and colon cancer, same as the extract. The synthetic trans-(±)- kusunokinin was found to suppress topoisomerase II, STAT3, CyclinD1 and p21 on breast cancer and cholangiocarcinoma cells”. Please correct this issue in the other paragraphs as well including the abstract. For example, line 1 “Trans-(-)-kusunokinin, an anticancer compound, partially binds CSF1R in breast cancer cells”. Regarding this last sentence, please remove “partially bind” everywhere in the text as it is highly misleading for the reader. A compound binds to a target (possibly with low affinity) or not.
Response: We replaced all words according to reviewer comment on lines 12, 72, 74, 76, 78, 80, 83, 85 and 87.
Q2. Other phrases should be rewritten as they remain unclear:
Q2.1 Line 99: replace “The docking score was represented by the relative binding energy (ΔGbind) in kcal/mol” by “The docking score was given as an effective binding energy (ΔGbind) in kcal/mol although it may differ from the true binding energy”.
Response: We replaced the sentence on page 3 lines 99-101 according to reviewer comment.
Q2.2 Line 111: “in order to circumvent the prediction of the relative binding energy”. The verb circumvent normally applies to a rule or a problem not a prediction. Please correct.
Response: We changed the word “circumvent” to “improve”. (Page 3, line 111)
Q2.3 Line 113: “under more realistic conditions”
Response: We changed “under more realistic conditions” to “with an effect of temperature, pressure and aqueous environment.” (Page 3, lines 113-114)
Q2.4 Paragraph from line 130: replace MD simulations by MD trajectories.
Response: We replaced the word on page 4 line 130 according to reviewer comment.
Q2.5 Line 141: “The RMSD plot denoted the stability of MD simulation after 50 ns of simulation time since the RMSD values in all 5 simulations constantly fluctuated (Figure 2A). Hence, taking the last 50 ns simulation was valid in terms of both structure and binding analysis.” Stability of a system is normally verified by checking that the RMSD plateaus soon after the beginning of the simulation with relatively low RMSD value (<3.0Å or so). Is it the case here? Also, it is not clear why Figure 2A does not start from 0 RMSD.
Response: We have verified the stability as you said. We start the simulation at the low RMSD. As the starting structure was energy-minimized, the starting RMSD was not zero, but in a rather low RMSD (<0.3 angstroms). The graph in the manuscript demonstrated RMSD from NPT simulation, which had been already equilibrated from the previous MD process. To make the reader clear, we have included the RMSD plot of the backbone atoms of NVT simulation in the supplementary material.
To make the content consistent, we also changed sentences on page 5 lines 141-146 to “The RMSD plot denoted the stability of MD simulation after 50 ns of NPT simulation time since the RMSD values in all 5 simulations constantly fluctuated (Figure 2A). Hence, taking the last 50 ns simulation was valid in terms of both structure and binding analysis. The plot of RMSD from the beginning of the simulation was shown in supplementary materials (Figure S1).” and added Figure S1 as supplementary materials.

Reviewer 2 Report
authors amended the manuscript according to the reviewer comments
Author Response
Thank you very much.
Reviewer 3 Report
The paper has improved and the corrections were addressed.
Few minor details.
Abstract. Last sentence. It could be better to change "bound HER2" to "may bind HER2"
Supplementary material: Change "Supplementary material of this" to "Supplementary material for this"
Author Response
Reviewer #3
Comments and Suggestions for Authors
The paper has improved and the corrections were addressed.
Few minor details.
Q1. Abstract. Last sentence. It could be better to change "bound HER2" to "may bind HER2"
Response: We changed last sentence on “Abstract” to “We conclude that trans-(±)-kusunokinin may bind HER2 with low affinity and had a different action from neratinib.”
Q2. Supplementary material: Change "Supplementary material of this" to "Supplementary material for this"
Response: We corrected the word according to reviewer comment on page 23 line 688.
